# LEARNING SO(3)-INVARIANT CORRESPONDENCE VIA POINT-WISE LOCAL SHAPE TRANSFORM

## ABSTRACT

Establishing accurate dense 3D correspondences between diverse shapes stands as a pivotal challenge with profound implications for computer vision and robotics. However, existing self-supervised methods for this problem assume perfect input shape alignment, restricting their real-world applicability. In this work, we introduce a novel self-supervised SO(3)-invariant 3D correspondence learner, dubbed LSTNet, that learns to establish dense correspondences between shapes even under challenging intra-class variations. Specifically, LSTNet learns to dynamically formulate an SO(3)-invariant local shape transform for each point, which maps the SO(3)-equivariant global shape descriptor of the input shape to a local shape descriptor. These local shape descriptors are provided as inputs to our decoder to facilitate point cloud self- and cross-reconstruction. Our proposed self-supervised training pipeline encourages semantically corresponding points from different shape instances to be mapped to similar local shape descriptors, enabling LSTNet to establish the dense point-wise correspondences. LSTNet demonstrates state-of-the-art performances on 3D semantic keypoint transfer and part segmentation label transfer given arbitrarily rotated point cloud pairs, outperforming existing methods by significant margins.

## 1 INTRODUCTION

Establishing dense 3D correspondences between different shapes is foundational to numerous applications across computer vision, graphics, and robotics (Saxena et al., 2006; Miller et al., 2003; Hao et al., 2013; Zeng et al., 2020). This task, however, remains challenging due to the high dimensionality and intricacy of 3D shape representations. One of the primary challenges hindering advancements in this domain is the difficulty of annotating dense inter-shape correspondences, which limits the leverage of strongly-supervised learning paradigms.

Recently, self-supervised learning methods have been proposed to address this issue (Liu & Liu, 2020; Cheng et al., 2021), showing promising directions for 3D correspondence estimation. Nonetheless, a significant limitation in current self-supervised learning approaches is their stringent assumption about the alignment of input shape pairs; these methods strongly assume that the input point cloud pair to establish correspondences between are precisely aligned. This assumption is rarely met in practice, where object scans and shape instances can be arbitrarily oriented. We find that the performance of existing methods degrades significantly when confronted with rotated input shapes, restricting their real-world applicability.

To address this challenge, we introduce a novel self-supervised learning approach, dubbed LSTNet, designed to reliably determine dense SO(3)-invariant correspondences between shapes via local shape transform (LST), irrespective of their rotational orientation. In essence, LSTNet learns to formulate SO(3)-invariant local shape transform for each point in a dynamic, input-dependent manner. Each point-wise local shape transform can map the SO(3)-equivariant *global* shape descriptor of the input shape to a *local* shape descriptor, which is passed to the decoder to reconstruct the shape and pose of the input point cloud. By training LSTNet in a self-supervised manner via self- and cross-reconstruction of input shapes, true semantically corresponding points are trained to yield similar local shape descriptors, enabling us to determine dense shape correspondences.

LSTNet demonstrates state-of-the-art performance on 3D semantic matching when evaluated on the KeypointNet dataset (You et al., 2020). In particular, significant improvements over existing base-

lines are observed when our method is applied to randomly oriented shape pair inputs. Furthermore, our approach also proves to be more effective compared to existing methods at part segmentation label transfer when evaluated on the ShapeNet dataset (Chang et al., 2015). This showcases not only the applicability of LSTNet across a diverse range of tasks but also its potential to be utilized for efficient dense annotation of 3D shapes. These results highlight the efficacy of LSTNet in addressing the challenges posed by real-world scenarios where existing methods fail to perform effectively.

The main contributions of our work can be summarized as follows:

- We introduce LSTNet, a novel self-supervised approach for determining dense SO(3)-invariant correspondences between arbitrarily aligned 3D objects.

- We propose to formulate the local shape information of each point as a novel function called *local shape transform* with dynamic input-dependent parameters, which effectively maps the global shape descriptor of input shapes to local shape descriptors.

- LSTNet achieves state-of-the-art performance on 3D keypoint transfer and part segmentation label transfer under arbitrary rotations, indicating its potential for application in a wide range of practical tasks in computer vision and beyond.

## 2 RELATED WORK

**Point cloud understanding via self-supervised learning.** While traditional methods for point cloud processing involving hand-crafted features (Tombari et al., 2010; Salti et al., 2014) have shown impressive performance, with the advent of deep learning, substantial research efforts have been directed towards developing learning-based algorithms capable of effectively processing and understanding point clouds (Qi et al., 2017a;b; Zhao et al., 2021; Choe et al., 2022). Due to limited large-scale datasets with rich annotations, self-supervised learning approaches emerged as a viable alternative. One of the most prominent directions to learn point cloud representations in an self-supervised manner is learning through self-reconstruction (Yang et al., 2018; Zhao et al., 2019; Pang et al., 2022) of the point cloud. Primarily inspired by the efficacy of point cloud reconstruction as a self-supervised representation learning scheme, we train LSTNet to establish 3D correspondences in a self-supervised manner via self- and cross-reconstruction of point clouds by leveraging SO(3)-invariant dynamic local shape transform.

**Equivariance and invariance to rotation.** The conventional method to improve a neural network's robustness to rotation is by employing rotation augmentations during training or inference. However, this tends to increase the resources required for training and still shows unsatisfactory results when confronted with an unseen rotation (Li et al., 2021a; Kim et al., 2023). In recent years, various methods have been proposed to yield point cloud representations which are equivariant (Cohen et al., 2018; Thomas et al., 2018; Shen et al., 2020; Chen et al., 2021) or invariant (Sun et al., 2019; Li et al., 2021b; Xiao & Wachs, 2021; Li et al., 2021a; Kim et al., 2023) to the rotation of the input, demonstrating enhanced performances under arbitrary input rotations. To facilitate the rotation-robust establishment of 3D dense correspondences, we utilize SO(3)-equivariant VNNs (Deng et al., 2021) in building LSTNet, leveraging its SO(3)-equivariant and -invariant representations to guarantee robustness to rotation by design.

**Establishing correspondences under intra-class variations.** Finding correspondences between images or shapes under intra-class variations - manifesting as differences in shape, size, and orientation within the same category of objects - poses significant challenges over photometric or viewpoint variations. This task has been widely studied in the domain of images, where existing methods make use of sparsely annotated image pair datasets to train their method in a strongly- or a weakly- supervised manner (Cho et al., 2021; Kim et al., 2022; Truong et al., 2022; Huang et al., 2022). However, learning to establish dense yet reliable 3D correspondences between 3D shapes remains challenging, as it is infeasible to label dense correspondence annotations across point cloud pairs with intra-class variations. Self-supervised methods have been proposed to address this issue (Liu & Liu, 2020; Cheng et al., 2021), but they strongly assume that the input point clouds are aligned, consequently leading to considerable significant degradation when confronted with arbitrarily rotated point clouds. To this end, we propose LSTNet to establish reliable 3D dense correspondences irrespective of the input point clouds' rotational orientation.

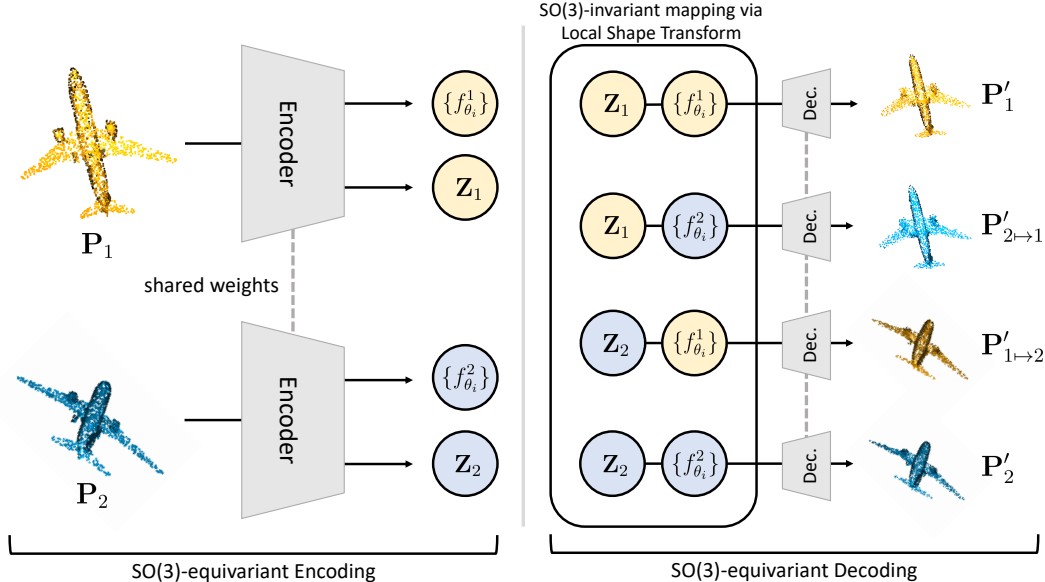

Figure 1: **Overview of self-supervised training of LSTNet**. The input point clouds are independently encoded to SO(3)-equivariant global shape descriptor $\mathbf{Z}$ and dynamic SO(3)-invariant point-wise local shape transforms $\{f_{\theta_i}\}$. The local shape transforms map the global shape descriptor to local shape descriptors by infusing local semantics and geometry, which are used as inputs to the decoder for self-reconstruction. For cross-reconstruction, we apply the local shape transforms formulated from *another* point cloud to reconstruct the point cloud, ensuring that the local shape descriptors successfully capture generalizable local semantics and geometries. We supervise LSTNet via penalizing errors in self- and cross- reconstructions. At inference, we can leverage the local shape transforms for obtaining local shape descriptors, to identify the dense correspondences.

## 3 LSTNet for 3D semantic correspondence

In this section, we introduce the components of LSTNet and their functionalities, which come together to facilitate the end-to-end self-supervised training for 3D semantic correspondence establishment. The objective of 3D semantic correspondence is as follows; given two different point clouds instances $\mathbf{P}_1 \in \mathbb{R}^{N \times 3}$ and $\mathbf{P}_2 \in \mathbb{R}^{N \times 3}$ belonging to the same semantic category, we aim to find all semantically corresponding point pairs $\{\mathbf{p}_i, \mathbf{q}_i\}_{i=1}^{N}$ such that $\mathbf{p}_i \in \mathbf{P}_1$ and $\mathbf{q}_i \in \mathbf{P}_2$. To achieve this, we claim it is crucial to identify the local shape information *i.e.*, local semantics and geometry, which is generalizable across different instances within the same category.

Therefore, the main idea of LSTNet is to dynamically generate an SO(3)-invariant local shape transform as a function for each point, such that each local shape transform can map the SO(3)-equivariant global shape descriptor of the input point cloud to its respective local shape descriptor. In the following, we elaborate on the network architecture of LSTNet, in particular how we leverage SO(3)-equivariant and invariant representations to facilitate the dynamic formulation of point-wise SO(3)-invariant local shape transforms and the reconstruction of pose-preserved point clouds (Sec 3.1). Subsequently, we introduce our self-supervisory objective function, which trains LSTNet to self- and cross-reconstruct the input point clouds in a rotation-equivariant manner (Sec 3.2), finally enabling the establishment of 3D dense correspondences (Sec 3.3) via corresponding local shape descriptors. Figure 1 illustrates the outline of the training scheme of LSTNet.

### 3.1 Network design of LSTNet

#### 3.1.1 Preliminary: SO(3)-equivariant Vector Neuron Networks

One of the main motivations of LSTNet is to establish reliable and accurate 3D dense correspondences given *arbitrarily rotated* shapes, a setting where existing work shows to be brittle. This

requires our encoder to formulate the point-wise local shape transforms not only effectively to capture the local shape semantics and geometry, but also robustly against transformations in the SO(3) space. To this end, we integrate Vector Neuron Networks (VNNs) (Deng et al., 2021) to function as the SO(3)-equivariant layers of our network architecture[1]. In VNNs, a single neuron, which is represented by a scalar-list of values, is lifted to a *vector-list* feature $\mathbf{V} \in \mathbb{R}^{C \times 3}$, which is essentially a sequence of 3D vectors. The layers of VNNs handle batches of such vector-list features such that equivariance with respect to rotation $R \in$ SO(3) is satisfied *i.e.*, $f(\mathbf{V}R) = f(\mathbf{V})R$. Not only can VNNs handle and yield SO(3)-equivariant representations, but they can also be used to obtain SO(3)-invariant features. Performing a product of an equivariant vector-list feature $\mathbf{V}R \in \mathbb{R}^{C \times 3}$ with the transpose of another consistently equivariant vector-list feature $\mathbf{U}R \in \mathbb{R}^{C' \times 3}$ yields an SO(3)-invariant output as follows: $(\mathbf{V}R)(\mathbf{U}R)^\top = \mathbf{V}RR^\top\mathbf{U}^\top = \mathbf{V}\mathbf{U}^\top$. We refer the readers to the original paper (Deng et al., 2021) for further information and detailed formulations.

### 3.1.2 SO(3)-EQUIVARIANT ENCODER

We design our encoder architecture to take as input a point cloud $\mathbf{P} \in \mathbb{R}^{N \times 3}$, and simultaneously output an SO(3)-*equivariant* global shape descriptor and formulate point-wise SO(3)-*invariant* local shape transforms.

**SO(3)-equivariant global shape descriptor.** Given a point cloud, we first aim to obtain the SO(3)-equivariant *global* shape descriptor $\mathbf{Z} \in \mathbb{R}^{C \times 3}$, which captures the pose and the global shape characteristics of the input point cloud. We leverage VN-DGCNN (Deng et al., 2021) as our encoder architecture, which consists of 4 edge convolutional VN-layers to capture local semantics at a progressively larger receptive field, and a FPN (Lin et al., 2017) to aggregate the multi-level features. Then, we apply the global average pooling to the aggregated SO(3)-equivariant point-wise features $\mathbf{V}^{\text{equi}} \in \mathbb{R}^{C \times 3 \times N}$ to encode SO(3)-equivariant global shape descriptor $\mathbf{Z}$ of the input point cloud. The global shape descriptor can be used subsequently as the input for our SO(3)-invariant point-wise local shape transform, to be mapped to their respective local shape descriptors, as shown in Figure 1.

**Dynamic SO(3)-invariant point-wise local shape transform.** Alongside the extraction of SO(3)-equivariant global shape descriptors, we also formulate the SO(3)-*invariant* local shape transform $f_{\theta_i} : \mathbb{R}^{C \times 3} \mapsto \mathbb{R}^{C' \times 3}$ for each point $\mathbf{p}_i \in \mathbb{R}^3$ of the input point cloud $\mathbf{P} \in \mathbb{R}^{N \times 3}$. The parameters of each local shape transform $\theta_i \in \mathbb{R}^{C' \times C}$ are input-dependent - thus, dynamic since they are predicted by our encoder for the $i$-th point of the point cloud. To predict $\theta_i$, we first obtain SO(3)-*invariant* point-wise features $\mathbf{V}^{\text{in}} \in \mathbb{R}^{C' \times 3 \times N}$ as described in Sec. 3.1.1. Then, we transform each vectorized SO(3)-invariant point-wise feature $\text{vec}(\mathbf{v}_i^{\text{in}}) \in \mathbb{R}^{3C'}$ to the vectorized parameter of the local shape transform $\text{vec}(\theta_i) \in \mathbb{R}^{C'C}$ by using a multi-layer perception. By reshaping $\text{vec}(\theta_i)$ to $\theta_i \in \mathbb{R}^{C' \times C}$, we finally obtain the dynamic and SO(3)-invariant local shape transform $f_{\theta_i}$ for the point $\mathbf{p}_i$. The role of these local shape transforms is to map the SO(3)-equivariant global shape descriptor $\mathbf{Z} \in \mathbb{R}^{C \times 3}$ to their respective local shape descriptors $\mathbf{v}_i' := f_{\theta_i}(\mathbf{Z}) \in \mathbb{R}^{C' \times 3}$, which is provided as the input to our decoder for reconstruction. Our self-supervised training scheme encourages the point-wise dynamic local shape transform to encapsulate the local shape information *e.g.*, semantics and geometry, to enhance the reconstruction performance.

### 3.1.3 SO(3)-EQUIVARIANT DECODER

Our decoder aims to reconstruct the initial input shapes using the obtained SO(3)-equivariant global shape descriptors $\mathbf{Z}$ and the SO(3)-invariant local shape transforms $\{f_{\theta_i}\}_{i=1}^N$. To reconstruct the point clouds aligned to their initial poses, we leverage SO(3)-equivariant layers as the building blocks of our decoder architecture. We first train our decoder to perform self-reconstruction, using the local shape descriptors $\mathbf{V}'$, *i.e.*, $\mathbf{P} \leftrightarrow \mathbf{P}' := \text{Decoder}(\mathbf{V}') = \text{Decoder}(\{f_{\theta_i}(\mathbf{Z})\}_{i=1}^N)$. We also train our decoder to perform cross-reconstruction, where we use the local shape descriptors obtained using global shape descriptors and local shape transforms from *different* point clouds. Specifically, assume we are given two point clouds $\mathbf{P}_1, \mathbf{P}_2 \in \mathbb{R}^{N \times 3}$, with SO(3)-equivariant global shape descriptors $\mathbf{Z}_1, \mathbf{Z}_2 \in \mathbb{R}^{C \times 3}$ and SO(3)-invariant local shape transforms $\{f_{\theta_i}^1\}_{i=1}^N, \{f_{\theta_i}^2\}_{i=1}^N$. We then can perform cross-reconstruction from $\mathbf{P}_1$ to $\mathbf{P}_2$ as follows: $\mathbf{P}_2 \leftrightarrow$

---

[1]Note that while any other SO(3)-equivariant network can be used in theory, we choose VNNs (Deng et al., 2021) for their simplicity and efficacy.

$\mathbf{P}'_{1\mapsto 2} := \text{Decoder}(\{f^1_{\theta_i}(\mathbf{Z_2})\}^N_{i=1})$. Intuitively, for the above cross-reconstruction to be carried out successfully, the local shape transforms for points of a *true* correspondence should hold similar dynamic parameters, mapping global shape descriptors to similar local shape descriptors. By training LSTNet to cross-reconstruct point clouds, we are supervising local shape transforms to map corresponding points between shapes to similar local shape descriptors, which encode local semantics and geometry that are generalizable across different instances within a category.

## 3.2 SELF-SUPERVISED OBJECTIVE

Due to the lack of annotated datasets for dense 3D inter-shape correspondences, we train LSTNet in a self-supervised manner by penalizing inaccurate shape reconstructions. First, we supervise LSTNet for self-reconstruction using the following loss:

$$\mathcal{L}_{\text{SR}} = \lambda_{\text{MSE}} \, \text{MSE}(\mathbf{P}, \mathbf{P}') + \lambda_{\text{EMD}} \, \text{EMD}(\mathbf{P}, \mathbf{P}'), \tag{1}$$

where MSE is the Mean Squared Error, EMD stands for the Earth Mover's Distance, and both $\lambda_{\text{MSE}}$ and $\lambda_{\text{EMD}}$ are weight coefficients. In essence, we are trying to minimize the difference between the input and reconstructed point cloud. We also supervise LSTNet for cross-reconstruction as follows: $\mathcal{L}_{\text{CR}} = \lambda_{\text{CD}} \, \text{CD}(\mathbf{P}_1, \mathbf{P}'_{2\mapsto 1})$, where CD stands for the Chamfer distance, and $\lambda_{\text{CD}}$ is a weight coefficient. Finally, our total loss $\mathcal{L}_{\text{total}}$ is defined as: $\mathcal{L}_{\text{total}} = \mathcal{L}_{\text{SR}} + \mathcal{L}_{\text{CR}}$. We omit the CD loss from self-reconstruction, as we can directly use the input point cloud to provide supervision using the MSE loss. We also omit the EMD loss from cross-reconstruction, as EMD tends to overlook the fidelity of detailed structures (Wu et al., 2021), which is crucial in cross-reconstruction of shapes under intra-class variations.

## 3.3 SO(3)-INVARIANT CORRESPONDENCE

In this section, given two randomly rotated point clouds $\mathbf{P}_1$ and $\mathbf{P}_2$, we elaborate on how our LSTNet establishes the 3D dense correspondence from $\mathbf{P}_1$ to $\mathbf{P}_2$. As shown in Figure 1, we first encode the SO(3)-equivariant global shape descriptor of $\mathbf{P}_2$, $\mathbf{Z}_2 \in \mathbb{R}^{C\times 3}$, and the SO(3)-invariant local shape transform functions of $\mathbf{P}_1$, $\{f^1_{\theta_i}\}^N_{i=1}$. Then, we cross-reconstruct $\mathbf{P}_2$ as follows: $\mathbf{P}'_{1\mapsto 2} := \text{Decoder}(\{f^1_{\theta_i}(\mathbf{Z_2})\}^N_{i=1})$. Finally, we define the 3D dense correspondence from $\mathbf{P}_1$ to $\mathbf{P}_2$ as the nearest point pairs among all possible pairs between $\mathbf{P}_2$ and $\mathbf{P}'_{1\mapsto 2}$. Since both encoder and decoder are SO(3)-equivariant, the cross-reconstructed point cloud $\mathbf{P}'_{1\mapsto 2}$ is aligned to $\mathbf{P}_2$. As a result, our LSTNet can predict 3D dense correspondences between randomly rotated point clouds, while previous approaches (Cheng et al., 2021; Liu & Liu, 2020) experience a high rate of failure.

## 4 EXPERIMENTS

We present evaluations of LSTNet on the tasks of 3D semantic keypoint transfer and part segmentation label transfer, following prior work (Liu & Liu, 2020; Cheng et al., 2021). We mainly evaluate under the I/SO(3) and SO(3)/SO(3) settings, where I/SO(3) refers to training with aligned shapes while testing on arbitrarily rotated inputs, and SO(3)/SO(3) uses arbitrarily rotated inputs for both training and testing. Note that Liu & Liu (2020); Cheng et al. (2021) both used the I/I settings, where the inputs were perfectly aligned even at test time - which is an unrealistic setting in practice.

**Datasets.** We use the KeypointNet dataset (You et al., 2020) to evaluate LSTNet on the task of 3D semantic correspondence. KeypointNet is a large-scale and diverse 3D keypoint dataset based on ShapeNet models, containing 103,450 keypoints and 8,234 3D models from 16 object categories. For the task of part segmentation label transfer, we use the ShapeNet part dataset (Chang et al., 2015). Following CPAE (Cheng et al., 2021), we use the same pre-processed ShapeNet part dataset provided by (Chen et al., 2019). For all datasets, we follow the original train/validation/test splits provided by the authors of ShapeNet, and generate all pairs of shapes (given $N$ shapes, $_N\mathbf{P}_2$ pairs) in the testing set as our test pairs. Note that we exclude shape pairs that do not share the same keypoint or part label to avoid interference from non-existing correspondences.

**Baseline methods.** Throughout the evaluation section, we mainly compare LSTNet against CPAE (Cheng et al., 2021), the state-of-the-art self-supervised method to establish 3D dense cor-

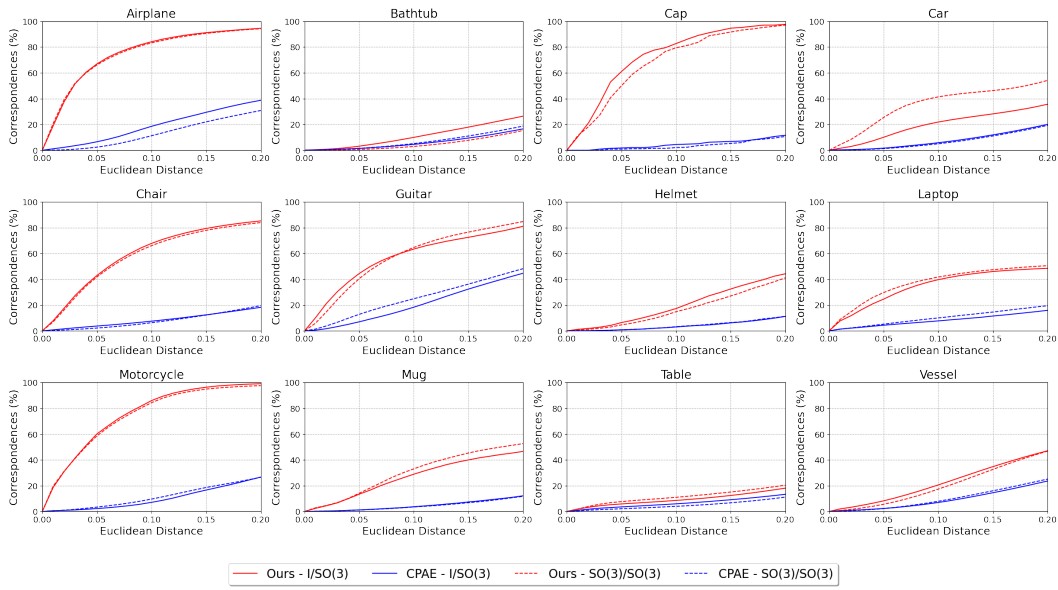

Figure 2: **Percentage of Correct Keypoints (PCK) for the 12 categories of KeypointNet dataset under varying thresholds under the I/SO(3) and SO(3)/SO(3) settings**. LSTNet consistently outperforms CPAE on all classes and thresholds on both evaluation settings.

respondence by exploiting an intermediate UV canonical space. When open-sourced pre-trained models are applicable, we also compare LSTNet with AtlasNetV2 (Deprelle et al., 2019), Fold-ingNet (Yang et al., 2018) and Liu & Liu (2020). AtlasNetV2 proposes to represent shapes as the deformation and combination of learnable elementary 3D structures, which can be extended to 3D correspondence establishment. FoldingNet introduces a folding-based decoder to to 'fold' a canon-ical 2D grid into the 3D object surface, where the canonical 2D grid can be applied to identify cross-shape correspondences. Liu & Liu (2020) formulates an implicit function to predict point-wise part embedding vectors a part embedding vector for each 3D point, which can also be used to establish correspondences.

**Implementation details.** We use VN-DGCNN (Deng et al., 2021) as our SO(3)-equivariant en-coder, and VN-based multi-layer perception as our SO(3)-equivariant decoder. For a fair compar-ison, we set the dimension, $C$, of SO(3)-equivariant global shape descriptor $\mathbf{Z} \in \mathbb{R}^{C \times 3}$ as 170 ($\approx 512/3$) since CPAE (Cheng et al., 2021) use 512-dimensional global shape descriptors. Follow-ing the training setup of CPAE (Cheng et al., 2021), we use $\lambda_{\mathrm{MSE}}$, $\lambda_{\mathrm{EMD}}$, and $\lambda_{\mathrm{CD}}$ as 1000, 1, and 10, respectively. We supervise LSTNet with self- and cross-reconstruction simultaneously in a single training stage, unlike previous methods (Liu & Liu, 2020; Cheng et al., 2021) which train their method with heuristic curriculum learning (*e.g.*, training self-reconstruction only for warm-up). LSTNet is implemented in PyTorch, and is optimized with the Adam (Kingma & Ba, 2014) optimizer at a constant learning rate of $1e^{-3}$.

### 4.1 3D SEMANTIC KEYPOINT TRANSFER

Following previous work (Cheng et al., 2021), we compute the distances from the transferred $M$ keypoints to the ground truth keypoints, and report PCK (Percentage of Correct Keypoints) of our transferred keypoints, which is computed by:

$$\mathrm{PCK} = \frac{1}{M} \sum_{m=1}^{M} \mathbb{1}[\|\mathbf{k}_m - \hat{\mathbf{k}}_m\| \leq \tau], \tag{2}$$

where $\tau$, $\mathbf{k}_m$, and $\hat{\mathbf{k}}$ is a distance threshold, $m$-th ground truth keypoint, and $m$-th transferred key-point. Therefore, a transferred keypoint is considered to be correct if its distance from the ground truth keypoint is within the distance threshold $\tau$. The results on the KeypointNet dataset are il-

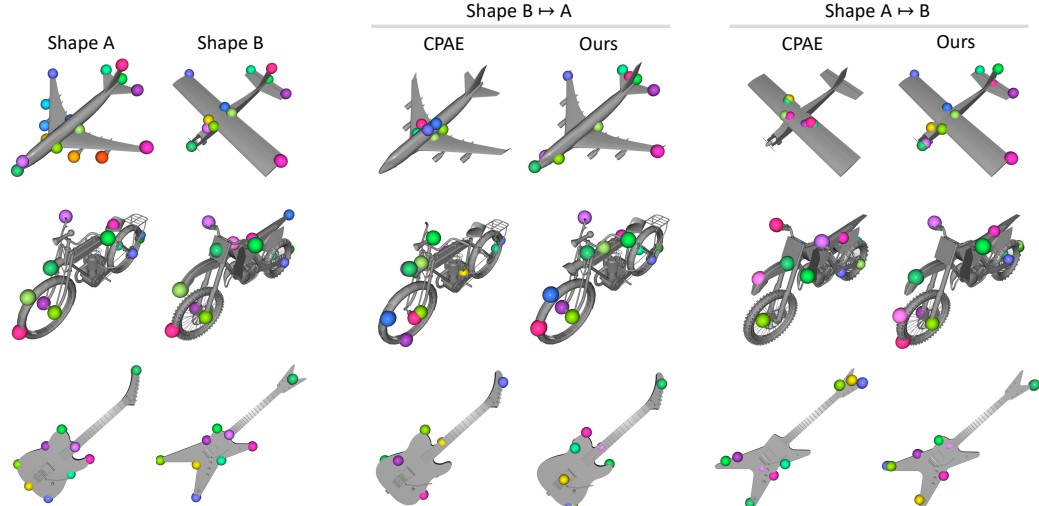

Figure 3: **Keypoint transfer results for airplane, motorcycle and guitar categories of Keypoint-Net**. Each row contains a shape pair, each with ground-truth keypoints and the keypoint transfer results. Note that the input shapes were arbitrarily rotated, but have been aligned for better visibility of keypoint transfer results. LSTNet shows to transfer the keypoints more accurately.

| Setting | Method | Airplane | Cap | Chair | Guitar | Laptop | Motorcycle | Mug | Table | Average |
|---------|--------|----------|-----|-------|--------|--------|------------|-----|-------|---------|
| I/SO(3) | CPAE (2021) | 21.0 | 38.0 | 26.0 | 22.7 | 34.9 | 14.7 | 51.4 | 35.5 | 30.5 |
| | LSTNet (ours) | 52.1 | 54.5 | 58.3 | 75.0 | 56.5 | 48.6 | 75.0 | 41.3 | **57.7** |
| SO(3)/SO(3) | CPAE (2021) | 17.0 | 36.6 | 24.5 | 39.4 | 37.4 | 15.8 | 51.9 | 36.7 | 32.4 |
| | LSTNet (ours) | 51.2 | 57.0 | 55.0 | 74.9 | 60.6 | 48.5 | 72.2 | 44.4 | **58.0** |

Table 1: **Average IoU (%) of part label transfer for eight categories in the ShapeNet part dataset.** We used the eight overlapping categories KeypointNet and ShapeNet datasets for evaluation. LSTNet outperforms CPAE on all classes on both evaluation settings.

lustrated in Figure 2 for varying distance thresholds $\tau$. It can be seen that LSTNet consistently outperforms CPAE on both settings of I/SO(3) and SO(3)/SO(3) for all classes, by up to 10x on certain classes and thresholds. This substantiates LSTNet's superior efficacy at establishing dense 3D correspondences between varying shapes. However, for certain classes such as Bathtub or Table, the performance is noticeably low, outperforming CPAE only by a tight margin. We speculate this to be due to the prevalent rotational symmetry of those classes, making it especially challenging to establish accurate 3D correspondences under arbitrary rotations. The qualitative results of LST-Net in comparison to baseline methods are presented in Figure 3. It can also be seen that LSTNet can identify more accurate keypoint correspondences compared to CPAE under arbitrary rotations, confirming the results presented in Figure 2.

## 4.2 PART SEGMENTATION LABEL TRANSFER

We evaluate LSTNet on the task of part segmentation label transfer on the ShapeNet part dataset (Chang et al., 2015), for the overlapping classes of KeypointNet and ShapeNet datasets. The qualitative results are presented in Table 1, where LSTNet outperforms CPAE on all classes by a large margin, by up to 3.3 times on the I/SO(3) setting, and 3.1 times on the SO(3)/SO(3) setting. We also provide the qualitative results of our part segmentation label transfer results in Figure 4. Attributing to the SO(3)-invariant nature of correspondences established by LSTNet, we are able to transfer part labels significantly more accurately given randomly rotated shape pairs.

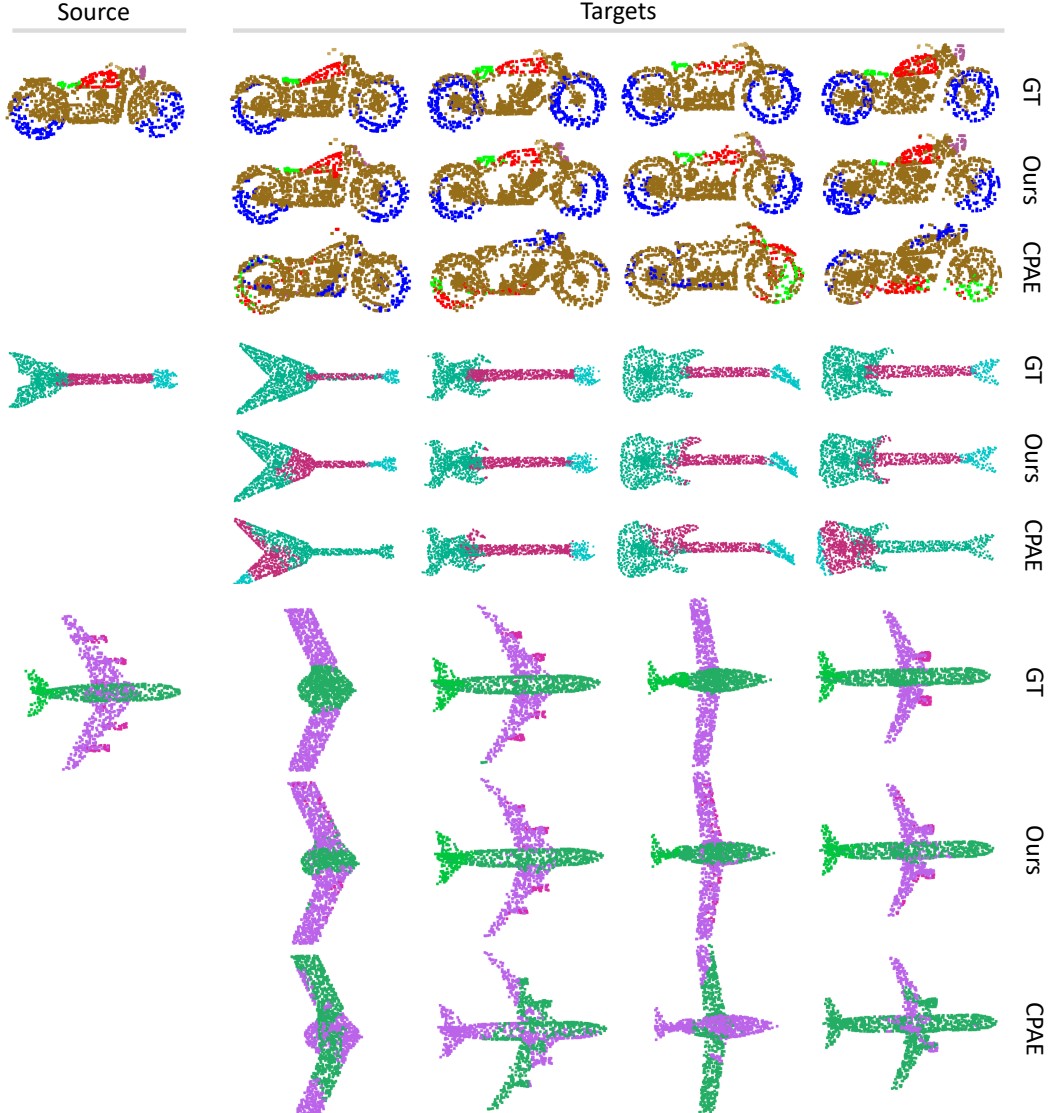

Figure 4: **Qualitative results of part label transfer on the ShapeNet part dataset.** The first row of the target indicates the ground truth instance part labels, while the rest shows the label transfer results via learned correspondences. Note that the input shapes were arbitrarily rotated, differently for each target column, but have been aligned for better visibility of part label transfer results. LSTNet shows to outperform CPAE (Cheng et al., 2021) consistently, showing high resemblance to ground truth results.

## 4.3 ABLATION STUDY AND ANALYSES

We perform an ablation study to justify the design choice of LSTNet, and evidence the efficacy of each component.

**Self- and Cross- reconstruction.** We train LSTNet in a self-supervised manner via penalizing errors in self- and cross- reconstruction of input point clouds. We conduct an ablation study on LSTNet's reconstruction, providing comparative results for scenarios with and without its use. The results are illustrated in the first graph of Figure 5. It can be seen that under both the I/SO(3) and SO(3)/SO(3) settings, incorporating *both* self- and cross- reconstruction yields the best results.

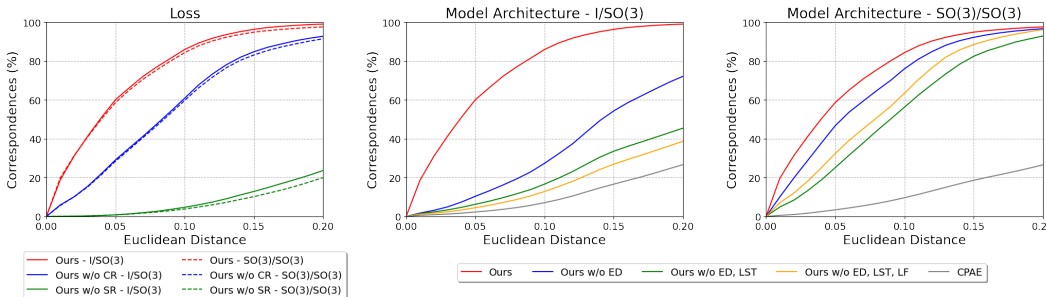

Figure 5: Ablation study on losses (the leftmost) and the components of model architecture (the others); self-reconstruction loss (SR), cross-reconstruction loss (CR), equivariant decoder (ED), local shape transform (LST), and local feature (LF). Omitting the equivariant decoder, local shape transforms, or local features defaults to using an SO(3)-decoder, UV coordinates (Cheng et al., 2021), or global features in encoding, respectively.

Removing self-reconstruction results in a much dramatic drop in performance; we conjecture this is because without self-reconstruction, the dynamic local shape transform fails to capture the required locality of its own point cloud in the first place, being unsuitable to establish correspondences.

**Encoder outputs and SO(3)-equivariance.** LSTNet uses VNNs as the SO(3)-equivariant layers to facilitate 3D dense correspondence establishment between arbitrarily rotated point cloud pairs, leveraging local shape transform to map global shape descriptors to local shape descriptors which encode the pointwise semantics and local geometry. We perform an ablation study to demonstrate the efficacy of local shape transforms and SO(3)-equivariant and -invariant representations in LST-Net on the motorcycle class of the KeypointNet dataset. We start our comparison from the architecture of CPAE (Cheng et al., 2021), given that they also employ an encoder-decoder architecture to self-supervise their network via shape reconstruction. The results are presented in the two rightmost graphs of Figure 5, showing the evaluation results under the I/SO(3) and SO(3)/SO(3) settings in order. It can be seen that our design choice of using equivariant encoders and decoders show consistent improvements over using an SO(3)-variant counterpart. Also, using UV coordinates as proposed in Deng et al. (2021) performs worse compared to our dynamic local shape transform, evidencing the comparatively better efficacy of transforming each point to their local shape descriptors via our dynamic SO(3)-invariant shape transform. While using local features as inputs to the encoder shows varied trends across I/SO(3) and SO(3)/SO(3) settings, using point-wise local feature as input is a key component which facilitates the learning of *point-wise* dynamic local shape transform that plays a critical role in establishing the 3D correspondences in LSTNet.

## 5 CONCLUSION

In this work, we introduced LSTNet, a novel self-supervised 3D semantic matching learner to identify dense SO(3)-invariant 3D correspondences between different shapes of the same semantic category even under arbitrary rotations. This robustness of LSTNet is facilitated by our novel use of SO(3)-equivariant and -invariant representations to formulate point-wise dynamic SO(3)-invariant local shape transform. Each local shape transform learns to map the global shape descriptor of a point cloud to a local shape descriptor while preserving SO(3)-equivariance, which can be used to reconstruct pose-preserved point clouds to finally establish dense SO(3)-invariant correspondences. The significant performance improvement over existing methods given rotated shapes on tasks of keypoint transfer and part label transfer broadens the applicability of 3D shape correspondences to various real-world tasks across computer vision and robotics *e.g.*, AR/VR and texture mapping. Furthermore, the superior ability of LSTNet to transfer part labels across shape instances can enhance the efficiency and accuracy of point-wise annotations of point clouds for various tasks. A promising direction for future research would be to promote the robustness and accuracy of dense 3D correspondences under real-world point cloud corruptions as well, such as occlusion or noise.

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

# A ADDITIONAL RESULTS ON KEYPOINTNET WITH I/I AND I/SO(3)

Figure 6: **Correspondence accuracy for 12 categories in the KeypointNet dataset.**

In this section, we provide the results of baseline methods (Deprelle et al., 2019; Yang et al., 2018; Liu & Liu, 2020; Cheng et al., 2021) and ours evaluated on the KeypointNet dataset, but under the I/I setting used by previous methods, as shown in Figure 6. Note that under the I/I setting, the input shape pairs are perfectly aligned both at train and test time - which is an unrealistic setting in practice. For CPAE Cheng et al. (2021) and ours, we also include the results under the I/SO(3) setting to visualize the performance difference between the two evaluation settings. It can be seen that while the drop in performance for CPAE from I/I to I/SO(3) setting is drastic, the difference is negligible in LSTNet, demonstrating the robustness of our SO(3) correspondence establishment scheme against arbitrary rotations. While LSTNet is not always competitive on all datasets, it is impractical to expect perfectly aligned shapes in real-world situations; on the realistic setting of SO(3) evaluation, LSTNet consistently shows the best results.

# B ADDITIONAL QUALITATIVE RESULTS.

We provide additional qualitative results which were not included in our manuscript due to space constraints, as shown in Figures 7, 8, and 9. It can be seen that under especially large intra-class variation, part segmentation label transfer using LSTNet often yields unsatisfactory results.

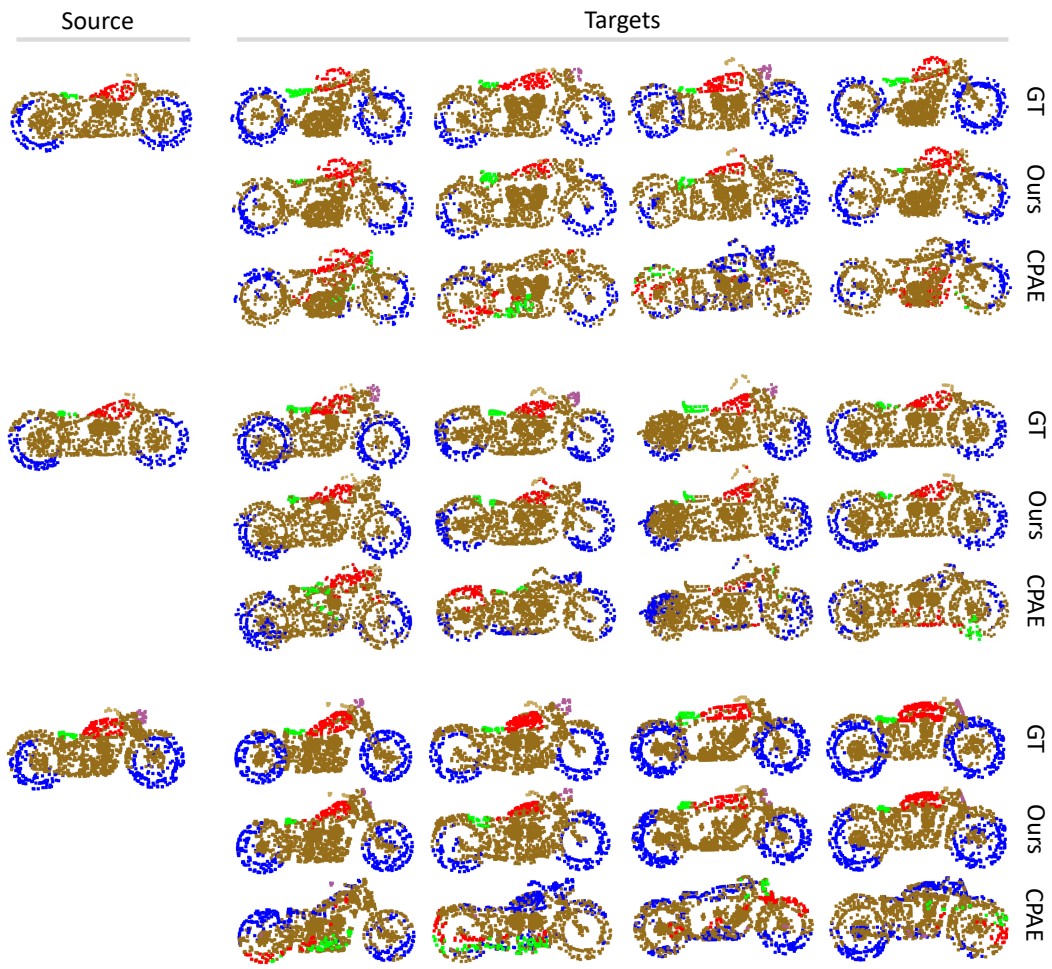

Figure 7: **Qualitative results of part label transfer on the motorcycle class of the ShapeNet part dataset.** Note that the input shapes were arbitrarily rotated, differently for each target column, but have been aligned for better visibility of part label transfer results. LSTNet shows to outperform CPAE Cheng et al. (2021) consistently, showing high resemblance to ground truth results.

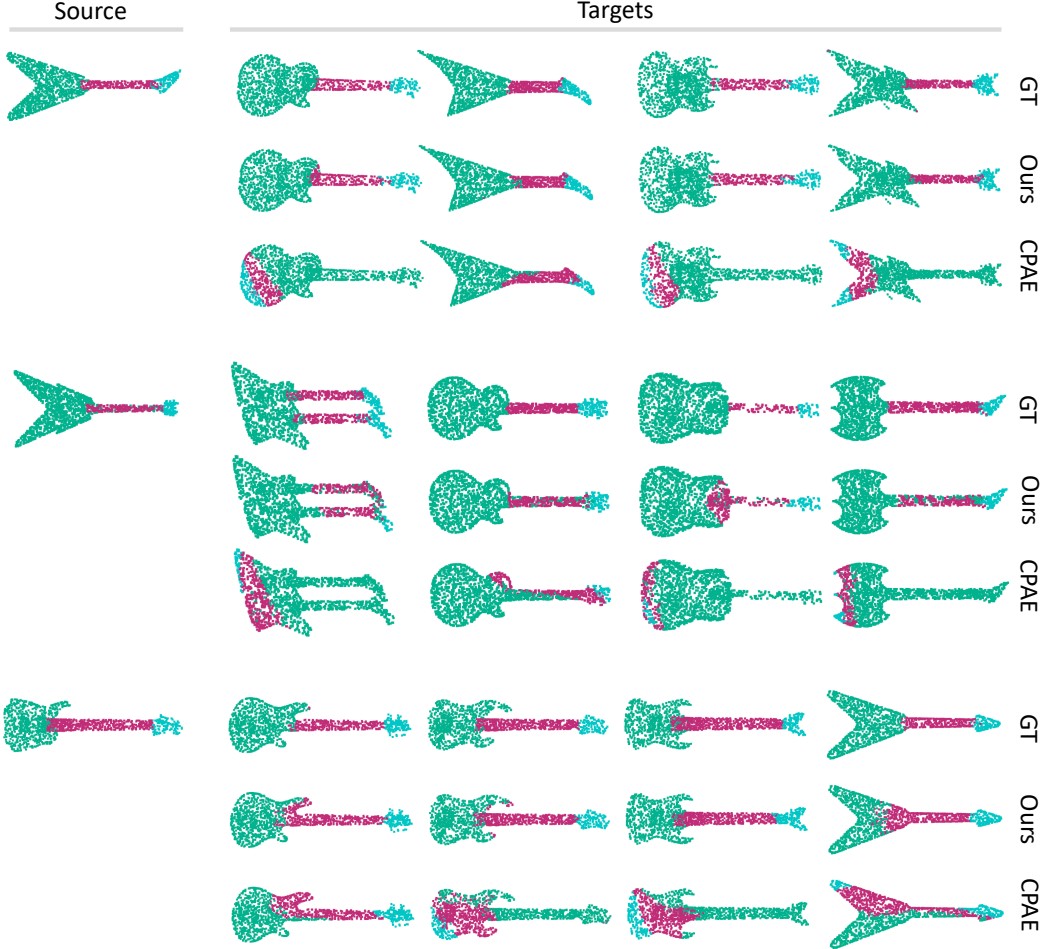

Figure 8: **Qualitative results of part label transfer on the guitar class of the ShapeNet part dataset.** Note that the input shapes were arbitrarily rotated, differently for each target column, but have been aligned for better visibility of part label transfer results. LSTNet shows to outperform CPAE Cheng et al. (2021) consistently, showing high resemblance to ground truth results.

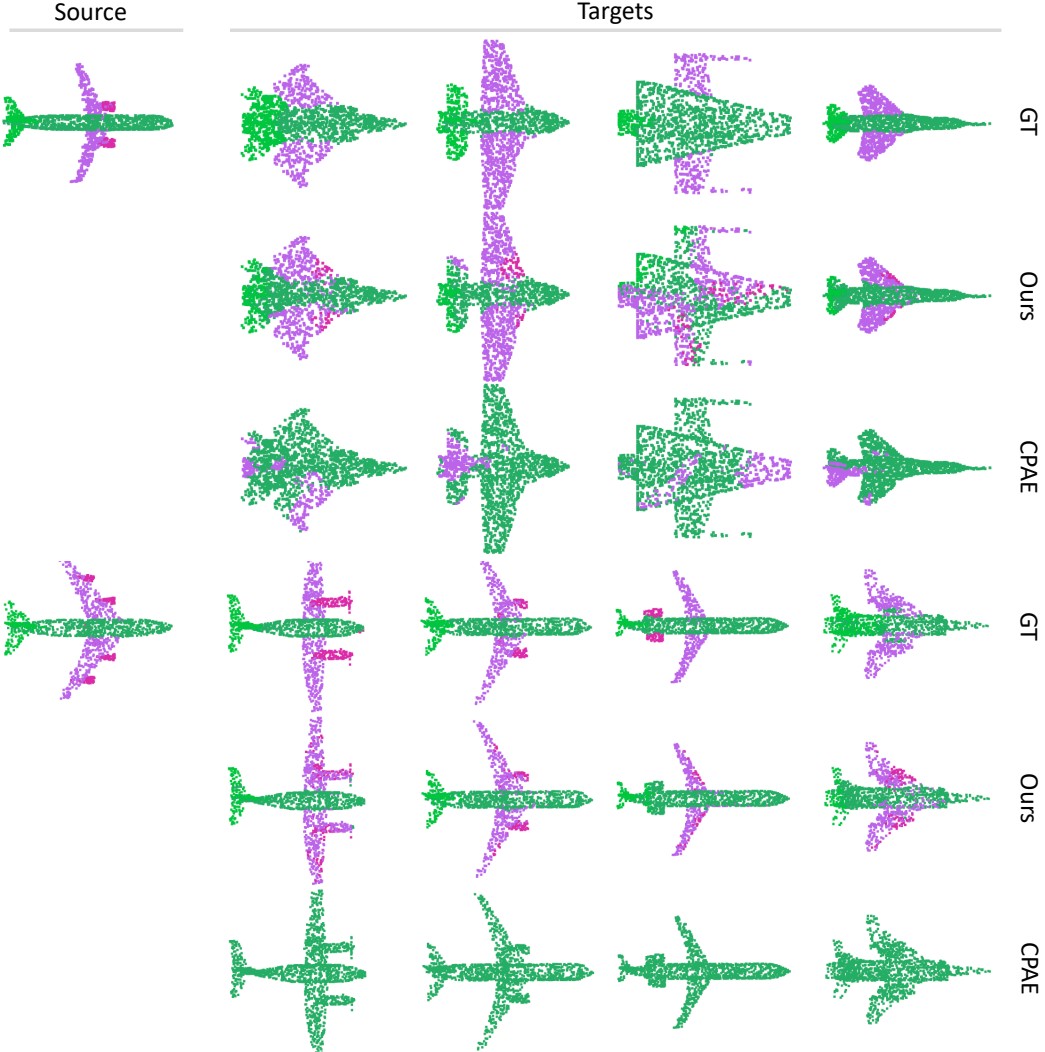

Figure 9: **Qualitative results of part label transfer on the airplane class of the ShapeNet part dataset.** Note that the input shapes were arbitrarily rotated, differently for each target column, but have been aligned for better visibility of part label transfer results. LSTNet shows to outperform CPAE Cheng et al. (2021) consistently, showing high resemblance to ground truth results.

