# OpenReview forum: "Learning SO(3)-Invariant Correspondence via Point-wise Local Shape Transform"
_ICLR.cc/2024/Conference — ICLR 2024 Conference Withdrawn Submission_

### Official Review · Reviewer_eS3u · 2023-10-25

**Soundness:** 3 good
**Presentation:** 2 fair
**Contribution:** 3 good
**Rating:** 6
**Confidence:** 2

**Summary:**

This work proposes LSTNet, a self-supervised method to establish reliable 3D dense correspondences irrespective of the input point clouds’ rotational orientation.

Specifically, LSTNet learns to formulate SO(3)-invariant local shape transform for each point in a dynamic, input-dependent manner. Each point-wise local shape transform can map the SO(3)-equivariant global shape descriptor of the input shape to a local shape descriptor, which is passed to the decoder to reconstruct the shape and pose of the input point cloud.

The proposed self-supervised training pipeline encourages semantically corresponding points from different shape instances to be mapped to similar local shape descriptors, enabling LSTNet to establish dense point-wise correspondences via nearest point pairs between cross-reconstructed point clouds.

**Strengths:**

The self- and cross-reconstruction training strategy is simple yet effective.

LSTNet demonstrates state-of-the-art performance on 3D semantic matching when evaluated on the KeypointNet dataset and part segmentation label transfer when evaluated on the ShapeNet dataset.

**Weaknesses:**

The performance of aligned shape pairs under the setting of I/I shows that other methods, such as CPAE, are much better than LSTNet.

**Questions:**

The reason why other methods are much better than LSTNet under the setting of I/I should be clarified.

Lack of limitations.

---

### Official Review · Reviewer_jP4i · 2023-10-30

**Soundness:** 3 good
**Presentation:** 3 good
**Contribution:** 2 fair
**Rating:** 5
**Confidence:** 4

**Summary:**

1) This paper proposes a self-supervised method to find semantically corresponding points for a point cloud pair;

2）The main idea is to decouple a point cloud feature learning process into a SO(3)-equivariant global shape descriptor and dynamic SO(3)-invariant point-wise local shape transforms;

3) Experiments on the KeypointNet dataset show the effectiveness of the proposed method.

**Strengths:**

1) This paper is generally well-written;

2) The idea of factorizing point cloud descriptors into SO(3)-equivariant global shape descriptor and dynamic SO(3)-invariant
point-wise local shape transforms seems to be novel;

3) Experimental results are good.

**Weaknesses:**

1) The main weakness of this paper could be all experiments are performed on synthetic datasets, with simple point cloud. It's good for authors' to show some examples/experiments on real-world datasets. For example, the 3Dmatch dataset.

2) Since the proposed method can estimate dense correspondences, I wonder whether the proposed method can be used to estimate the relative rotation/translation for a point cloud pair. For example, the estimated dense correspondences can be fed to an ICP method to estimate the relative rotation/translation.

3) The running time and GPU memory cost is blurry for me;

4) Please compare the proposed method with more recent papers, e.g., [SC3K: Self-supervised and Coherent 3D Keypoints Estimation
from Rotated, Noisy, and Decimated Point Cloud Data].

**Questions:**

Please refer to the weaknesses.

---

### Official Review · Reviewer_wiS9 · 2023-10-30

**Soundness:** 2 fair
**Presentation:** 2 fair
**Contribution:** 2 fair
**Rating:** 3
**Confidence:** 4

**Summary:**

This paper introduces LSTNet, which leverages an SO(3)-equivariant encoder-decoder architecture(Vector Neuron Networks, VNNs) and proposes a novel function called local shape transform to further transform the learned features. The proposed method is validated on both the 3D keypoint transfer and part segmentation label transformer tasks.

**Strengths:**

1. The idea of cross-reconstruction for generating inter-object correspondences in a self-supervised way is interesting;

2. The overall writing is good and the methodology part is well-organized and easy to follow.

**Weaknesses:**

1. The novelty of this work seems insufficient for ICLR. The whole pipeline heavily relies on VNNs and the main contribution I personally consider is the local shape transform and the self-supervised mechanism for correspondences.

2. Regarding the local shape transform:
   2.1. From 3.1.1, the SO(3)-invariant output is $\mathbf{V}\mathbf{U}^T \in \mathbb{R}^{C \times C}$, while in 3.1.2, the obtained SO(3)-invariant features $\mathbf{V} \in \mathbb{R}^{C^\prime \times 3 \times N}$ have a different shape;

   2.2 The authors claimed that the local shape transform transforms the global features to local ones. Regarding this, I have two questions.

      2.2.1 First, why are the features obtained by the Encoder global? They are generated by a DGCNN-based VNN, but DGCNN is not guaranteed to capture the global context, as it is graph-based and really depends on the number of layers together with the number of rings of each layer.

      2.2.2 Second, the so-called local shape transform is predicted by a multi-layer perception from some SO(3)-invariant features that obtained from the input. Why after transforming the "global" features by such a mechanism, the features turn to "local"? I cannot see any specific design that enables it. It should be further explained. (I personally do not think so)

3. Regarding the experiments:
    3.1 The experiments are only conducted on synthetic data, which cannot support the proposed method can work for real applications. I think it would be better to have additional real-data experiments;

     3.2 As this paper also targets on correspondence estimation, whose typical downstream task is pose estimation. Therefore, I consider it worthwhile to also conduct experiments on tasks of 6D pose estimation or point cloud registration (there you always use real data), to further validate the estimated correspondences.

    3.3 In Tab.1, only CPAE proposed in 2021 is used as the baseline. Some recent methods, e.g., [1], should also be included. Otherwise the results are not convincing at all (only compared to a single baseline which was proposed years ago). And it seems CPAE is the only baseline method for all the experiments. More baselines are required on both tasks.

   3.4 The method is claimed to generate SO(3)-invariant correspondences. However, in Tab. 1, even on the synthetic data, the I/SO(3) and SO(3)/SO(3) experiments perform unsimilarly (I would expect to have similar results per category, as it is on synthetic and clean data). Could this be explained?

4. For the SO(3)-equivariant and -invariant methods, some works for point cloud registration [2, 3, 4, 5] should also be discussed.
---------------------------------------------
[1]. Zohaib et al. SC3K: Self-supervised and Coherent 3D Keypoints Estimation from Rotated, Noisy, and Decimated Point Cloud Data, ICCV 2023;

[2]. Dent et al. PPF-FoldNet: Unsupervised Learning of Rotation Invariant 3D Local Descriptors, ECCV 2018

[3]. Ao et al. SpinNet: Learning a General Surface Descriptor for 3D Point Cloud Registration, CVPR 2021

[4]. Wang et al. You Only Hypothesize Once: Point Cloud Registration with Rotation-equivariant Descriptors, ACM MM 2022

[5]. Yu et al. Rotation-Invariant Transformer for Point Cloud Matching, CVPR 2023

**Questions:**

See weaknesses.

---

### Official Review · Reviewer_a6Ps · 2023-10-31

**Soundness:** 3 good
**Presentation:** 3 good
**Contribution:** 3 good
**Rating:** 5
**Confidence:** 4

**Summary:**

This paper attempts to register point cloud properties to their templates without precise correspondences and exact shape matching. To achieve this, the authors trained a local shape transform (LST) network that produces SO(3) invariant correspondences. The training is self-supervised. The experimental results on ShapeNet look nice.

**Strengths:**

- Valid motivation. Unlike the abused topic, vanilla point cloud registration, the motivation stands and could potentially benefit practical usages.
- The SO(3)-invariant network design intrinsically ensures robustness against rotations.
- The joint usage of a global descriptor and a local descriptor makes sense and may help with classification and recognition directly.
- The self-supervision scheme looks plausible by self and cross-reconstruction.

**Weaknesses:**

My major concern is with the experimental setup. While the experiments on ShapeNet is common in the community and shows good result, I am in general doubtful whether such an approach could be really applied to the real world.
In motivation, the authors talk about usage in vision, graphics, and robotics. In vision and robotics, we are interested in fitting real-world scans to templates (e.g. [Scan2CAD, CVPR 2019]), where in most cases, only noisy, partial, and sparse point clouds are provided. The authors do not have experiments or discussions in such cases.

The authors also take groundtruth keypoints and semantic segmentations from datasets for the experiments. In the real-world, however, obtaining such accurate high-level semantic information already requires a deep understanding of the point cloud, and its segmentation backbone may already be SO(3) invariant. This impairs the strength that the authors proposed.

**Questions:**

Following my points in the "weaknesses" section, I am curious about several relevant problems in the practical setup (i.e., scan to model).
1. Would SO(3) invariance be sufficient? Do we need SE(3) or even Sim(3) invariance, if we cannot easily normalize the input due to the noise and sparsity?
2. Will the network still be functional if the density distributions are different across input and output?
3. Will it work out of the 16-category domain? Do we need more training data, or would it work out-of-box?
4. Would non-gt and/or biased key points and semantic parts be transferred properly?

It would be nice if the authors could conduct a minimal set of experiments in the real-world setup (e.g., extract a reconstruction from a ScanNet model and attempt to apply keypoint/semantic part transfer). Otherwise, it would be good to see a justification that this paper itself is an inevitable intermediate step toward real-world usage, and what can be done to further extend it.

---

### Official Review · Reviewer_Frem · 2023-11-07

**Soundness:** 3 good
**Presentation:** 3 good
**Contribution:** 2 fair
**Rating:** 5
**Confidence:** 4

**Summary:**

This paper presents a method of learning dense 3D correspondence between shapes in a self-supervised manner. Specifically, it is built on an existing SO(3)-equivariant representation. The input point clouds are independently encoded to SO(3)-equivariant global shape descriptor Z and dynamic SO(3)-invariant point-wise local shape transforms. Then the network is trained via penalizing errors in self- and cross- reconstructions via the decoder. The experiment validates the effectiveness of the proposed method.

**Strengths:**

1. The paper is in general well organized and easy to follow.
2. The proposed method is straightforward and shown to be effective on the test data.

**Weaknesses:**

1. The main issue of the proposed method lies in the experimental evaluation. Only one learned-based method is adopted for comparison in the main paper on a rather simple dataset. More methods including some traditional methods should be also evaluated for better comparison. The experiment on the real dataset should be also provided to show the robustness of the proposed method.
2. From Fig. 6 in the supplementary, we can see that the performance of the proposed method on the I/I scenario is much worse than the SOTA method. More analysis of the drop of performance should be given. Moreover, the performance of different methods with different rotation angles should be provided for better comparison.
3. How about the performance of other methods with a rough alignment of the initial shape? If a rough alignment is enough for the existing methods, why should we learn SO(3)-invariant correspondence in an end-to-end manner?
4. The whole method is mainly built upon the existing SO(3)-equivariant representation. The main contribution lies in introducing this representation to the specific task. I didn't get too much novel insight in terms of network design.

**Questions:**

Please refer to the Weaknees part.

---

### Author Response · Authors · 2023-11-14
**Response to all reviewers**

We appreciate all the constructive comments from the reviewers (eS3u, jP4i, wiS9, a6Ps, Frem). We will do our best to reflect each of them, and improve the organization and the presentation of our paper. We will revise our paper by reflecting on the comments and resubmit it to another venue.